MicroRNA-99 family in cancer: molecular mechanisms for clinical applications

Wang Yueyuan 1
Huang Dan 1
Li Mingxi 2
Yang Ming 1 yangming@jlu.edu.cn
1 Department of Breast Surgery, General Surgery Center, The First Hospital of Jilin University , ChangChun, Jilin , China
2 Department of Neurology and Neuroscience Center, The First Hospital of Jilin University , ChangChun, Jilin , China
Haraguchi Tokuko
Electronic publication date: 2025 Mar 27
Publication date: 2025
Volume: 13
Electronic Location ID: e19188
Received 2024 Sep 16; Accepted 2025 Feb 25
Copyright: © 2025 Wang et al.
Copyright year: 2025
Copyright holder: Wang et al.
License: This is an open access article distributed under the terms of the Creative Commons Attribution License, which permits unrestricted use, distribution, reproduction and adaptation in any medium and for any purpose provided that it is properly attributed. For attribution, the original author(s), title, publication source (PeerJ) and either DOI or URL of the article must be cited.
License URL: https://creativecommons.org/licenses/by/4.0/

Keywords: miR-99, Cancer, Resistance, Biomarker

Funding: Wu Jieping Medical Foundation 320.6750.2023-3-36 This research was funded by Wu Jieping Medical Foundation, grant number: 320.6750.2023-3-36. The funders had no role in study design, data collection and analysis, decision to publish, or preparation of the manuscript.

==============================
MicroRNAs (miRNAs) are a class of non-coding RNA sequences that regulate gene expression post-transcriptionally. The miR-99 family, which is highly evolutionarily conserved, comprises three homologs: miR-99a, miR-99b, and miR-100. Its members are under-expressed in most cancerous tissues, suggesting their cancer-repressing properties in multiple cancers; however, in some contexts, they also promote malignant lesion progression. MiR-99 family members target numerous genes involved in various tumor-related processes such as tumorigenesis, proliferation, cell-cycle regulation, apoptosis, invasion, and metastasis. We review the recent research on this family, summarize its implications in cancer, and explore its potential as a biomarker and cancer therapeutic target. This review contributes to the clinical translation of the miR-99 family members.

Introduction

MicroRNAs (miRNAs) are endogenous, non-coding, single-stranded RNAs 18–25 nt in length. They recognize their targets via complementary binding between the “seed sequence”—an approximately 2–7 nt region at the 5′ end of the mature miRNAs, and a preferentially conserved site within the 3′-untranslated region of the mRNA (Friedman et al., 2009). Within the canonical machinery, miRNAs only degrade the mRNA of a target gene when entirely complementary. However, endogenously expressed miRNAs are typically not entirely complementary to their targets; they suppress target expression by inhibiting translation without affecting mRNA stability. Within the non-canonical machinery, miRNAs activate target mRNA translation directly or by binding to a conserved AU-rich sequence (Liu et al., 2023; Vasudevan, Tong & Steitz, 2007). miRNA targets more than 5,300 human genes, which represented 30% of our gene set, and are involved in various processes, such as cell proliferation, differentiation, apoptosis, autophagy, immune responses, metabolic homeostasis, and tumorigenesis (Kloosterman & Plasterk, 2006; Lewis, Burge & Bartel, 2005). Dysregulation of miRNAs has been increasingly detected in almost all types of cancer, indicating that miRNAs are pivotal factors in carcinogenesis (Kim & Croce, 2023).

The global cancer burden is increasing yearly, impacting millions of people and their families annually. Data from the International Agency for Research on Cancer indicate that in 2022, nearly 20 million new cancer cases emerged, accompanied by 9.7 million cancer-related deaths (Bray et al., 2024). Combating the cancer burden demands a comprehensive strategy that takes into account the drawbacks of existing diagnostic and treatment approaches. Present diagnostic methods, like imaging and biopsy, frequently have deficiencies in terms of specificity and sensitivity. For example, conventional imaging might fail to detect early-stage tumors or yield false positives, causing unwarranted distress and invasive procedures for patients. Additionally, established treatment methods, such as chemotherapy and radiotherapy, can be constrained by their side effects and the emergence of resistance. This situation highlights the importance of continuous research on potential biomarkers. These biomarkers can enable earlier cancer detection, offer more accurate prognostic details, and facilitate the development of more personalized treatment plans.

Genes encoding microRNAs are abundant in the genome. Depending on their origin, miRNAs can be divided into intronic, intergenic, and polycistronic miRNAs (within the host gene). Members of the miR-99 family are located close to members of the let-7 and miR-125 families, with which they form evolutionarily conserved clusters (Roush & Slack, 2008). Given that let-7 was the first recognized miRNA and that the miR-125 family is involved in leukemia development and myeloid activation, the roles of these families in tumorigenesis and cancer progression have been described (Feng et al., 2014; Roush & Slack, 2008; Sun, Lin & Chen, 2013; Wang, Wang & Li, 2019). However, few studies have examined the role of miR-99 family members in cancer. Eniafe & Jiang (2021) determined the functional roles of miR-99 family members in cancer and immunity, aiming to elucidate the multifaceted regulatory molecular mechanisms of miR-99 family members in biological processes. However, the potential clinical applications of the miR-99 family members in cancer diagnosis, prognosis, and treatment have not yet been thoroughly reviewed. We generated insights into the roles of miR-99 family members in cancer molecular regulation as well as novel research on the clinical translation of the miR-99 family in cancer.

Survey methodology

A literature search was conducted using PubMed and the Web of Science. The keywords utilized include “miR-99”, “miR-99a”, “miR-99a-3p”, “miR-99a-5p”, “miR-99b”, “miR-99b-3p”, “miR-99b-5p”, “miR-100”, “miR-100-3p”, “miR-100-5p” and “cancer”. The final selected references included studies on the expression of miR-99 family members in various cancers, anticancer therapeutic strategies targeting miR-99 family members, and miR-99 family potential clinical applications. The search for articles was not refined by publication date, authors, or author affiliations. We searched the literature describing the biogenesis of miRNAs and the probable mechanisms that regulate miR-99 family expression. After removing duplicate or irrelevant articles, 261 were selected for inclusion in this review (Fig. S1).

Genomic organization of mir-99 family

miRNA families are groups of homologous genes with highly similar seed sequences that encode different mature sequences (O’Brien et al., 2018). The miR-99 family includes three homologs, miR-99a, miR-99b, and miR-100, which are encoded on chromosomes 21, 19, and 11, respectively, and are modulated by different host genes. According to miRBase, the significant enhancement of the transcription at the 5′ ends of the miR-99 precursor is highly conserved in vertebrates and invertebrates; furthermore, the mature sequence of miR-99a-5p differs from that of miR-99b-5p by 1 nt and from that of miR-100-5p by 4 nt, although they share the same 2–8 nt seed region (Houbaviy, Murray & Sharp, 2003; Landgraf et al., 2007; Lui et al., 2007).

In humans, miR-99a, miR-125b-2, and let-7c form a cluster in MIR99AHG; miR-99a is encoded 658 bp upstream, and miR-125b-2 is encoded 50 kb downstream of let-7c. SPACA6R-AS hosts the miR-99b/let-7e/miR-125a cluster in which all three miRNAs are close to each other (<1 kb apart) (Cerami et al., 2012). The cluster comprising the homolog miR-100, let-7a-2 and miR-125b-1 is hosted by MIR100HG; miR-100 is located 5.7 kb upstream and miR-125b-1 is located 47 kb downstream of let-7a-2 (Lu et al., 2017) (Fig.1B). Although the distances between let-7c and miR-125b-2 and between let-7a-2 and miR-125b-1 exceed the 10 kb standard intergenic distance for miRNA clusters, genomic analysis shows that this pattern is conserved in various species (Christodoulou et al., 2010; Roush & Slack, 2008). Studies have verified the polycistronic nature of the miRNAs in these clusters, with the ca. 125 bp miR-99a/-100 region representing a cluster (Emmrich et al., 2014a; Lu et al., 2017).

Figure 1 Regulation of expression of miR-99 family members.

Diagram depicts some of the identified mechanisms of regulation of miR-99 expression. Transcriptional regulation can be characterized at six aspects: (i) Regulation of miR-99 host genes promoters. TLX3, ZEB1, FOXA1, HOXA10, ELK1, SMAD2/3 act as the activators of the miR-99 host genes transcription, whereas GATA6 and AR perform repressive function. TGF-β triggers MIR100HG transcription by SMAD2/3. CI-4AS-1, a kind of AR agonists, exerts inhibited functions on MIR99AHG enhancer by recruiting the histone methyltransferase EZH2. (ii) Regulation of miR-99 promoters. Vitamin D3 induces miR-99b expression by activating its receptor VDR to bind to the promoter domain of miR-99b. LPS promotes NF-κB to translocate the miR-99a and miR-100 promoter regions to increase the corresponding miRNA transcription. (iii) Epigenetic regulation of the miR-99 family. (iv) METTL14 increases pri-miR-99a expression via m6A modification. (v) The exon 1-3 of NRIP1 fuses with the exon 7-9 of MIR99HG elevates miR-99a transcripts (gene fusion), and (vi) several single nucleotide polymorphism (SNPs) and rare mutations within pri-miRNA sequences have been reported (miRNAs polymorphism). Post transcriptional regulation can also be achieved at two characterized levels: (vii) mature miRNAs are sponged by either circular or linear ncRNAs. Hypoxia and insulin treatment facilitate HIF-1α to bind to its response elements on RAETIK promoter to activate the expression of this lncRNA, turning to decrease miR-100-5p levels. (viii) Editing of miRNA sequences which interferes with mRNA target specificity as well as miRNA expression.

Regulation of mir-99 expression

Because miR-99 family members are encoded by different host genes, multiple members can be expressed as a single polycistronic transcript under the regulation of the host gene promoter (Baskerville & Bartel, 2005). Both T leukemia homeobox 3 (TLX3) and androgen receptor bind to MIR99AHG, subsequently modulate the expression of miR-99a and miR-125b (Renou et al., 2017; Sun et al., 2014). Knockdown of ZEB1, the promoter of the miR-99b/let-7e/miR-125a cluster, reduces both mature and primary miRNA expression (Ma et al., 2017). GATA6 represses MIR100HG promoter activity, whereas FOXA1, SMAD2/3 and ELK1 enhance it (Lu et al., 2017; Ottaviani et al., 2018; Su et al., 2019; Xu et al., 2021). HOXA10 transcribes the tricistron composed of miR-99a, miR-100, and miR-125b, as well as individual miRNAs in these clusters (Emmrich et al., 2014a, 2014b; Lu et al., 2017).

By contrast, miRNAs possess their own promoters and can be transcribed independently of their host genes. The miR-100 promoter can be directly activated by the transcription factors FOXA1, C/EBPα, ZEB1, and NME2 (Chen et al., 2014; Gong et al., 2020; Peng et al., 2020; Shi et al., 2015) and repressed by the stemness factors NANOG, OCT4, and SOX2 (Seol et al., 2020). STAT1 and vitamin D receptor (VDR) bind to the promoter domain of miR-99b to induce its expression of miR-99b and pri-miR-99b (Chang et al., 2019; Du et al., 2024). Lipopolysaccharide (LPS) treatment promoted NF-κB nuclear translocation, which in turn positioned the miR-99a and miR-100 promoter regions to increase the corresponding miRNA transcription (Bao et al., 2016; Jeon et al., 2015). The melanoma master transcription regulator MITF binds to the promoters of the miR-99a/let-7c/-125b-2 cluster and recruits TRIM28 to the miR-99a and miR-125b-2 regions, thereby inhibiting RNA polymerase II activity and attenuating its production (Sheinboim et al., 2021). Some regulators modulate miRNA transcription without directly interacting with promoters of the miR-99 family. EZH2 recognizes the let-7 promoter and inhibits the transcription of miR-99a and let-7c (Wu et al., 2023). In situ interaction analysis has revealed that the interaction between 5-lipoxygenase (5-LO) and Dicer downregulates the processing of the let-7e precursor, increasing the levels of miR-125a and miR-99b levels (Uebbing et al., 2021). IGFBP6 binds to insulin-like growth factor (IGF) and prevents its interaction with receptors, whereas the silencing of IGFBP6 increases the gene expression of miR-100 and let-7a-2 (Poloznikov et al., 2019). Myc represses the transcription of miR-99a and miR-125b (Chang et al., 2008). In kidney cancer, the expression of miR-100 is downregulated by PTEN, whereas in breast cancer (BC), it is upregulated by EphB6 (Bhushan & Kandpal, 2011; Majewska et al., 2022). In BC, BRCA1 induces transcription of miR-99a and miR-99b (Tanic et al., 2012).

Similar to coding genes, the expression of miRNAs is also epigenetically regulated. Yin Yang 1 (YY1) recruits HDAC5 to the miR-99a promoter and subsequently enhances the deacetylation of miR-99a to attenuate its expression (Qian, Wang & Li, 2020). PRMT5 represses miR-99 family transcription through symmetrical dimethylation of histone H4R3 in its promoter region (Jing et al., 2018). Interestingly, epigenetic regulation of the miR-99 family is invariably not repressive. For instance, in esophageal squamous cell carcinoma (ESCC), METTL14 upregulates miR-99a-5p by modulating the processing of m6A-mediated DGCR8-dependent pri-miR-99a (Liu et al., 2021b). Additionally, linear non-coding RNAs (ncRNAs) NCK1-AS1 methylates and reduces the transcription of the miR-100 precursor (Le et al., 2020).

Competing endogenous RNAs (CeRNAs) regulate miRNA transcription by interacting with miRNAs. Both linear and circular ncRNAs can inhibit miRNA function by binding to complementary sequences, thus preventing the interaction between miRNAs and their target mRNAs in a process known as “sponging” (Chan & Tay, 2018). circMCTP2 and circGFRA1 reportedly sponge miR-99a, whereas circ_0072309, circ_0006168, and circCASC15 sponge miR-100 (Cao et al., 2021; Shi et al., 2019; Sun et al., 2020; Yao et al., 2022; Yuan et al., 2022). The lncRNAs HAGLROS, SDCBP2-AS1, and RAETIK competitively sponge miR-100-5p (Chen et al., 2018b; Li et al., 2021; Liu et al., 2021a; Shu et al., 2022; Zhou et al., 2020). In nasopharyngeal carcinoma (NPC), the passenger strand of miR-100, miR-100-3p, is adsorbed by lncRNA ZFAS1 (Peng et al., 2022). LINC00589 functions as a ceRNA, simultaneously sponging miR-100 and releasing downstream target repression (Bai et al., 2022). Sponging by the lncRNAs UCA1 and DLEU1 restricts the functions of miR-99b-3p and miR-99b-5p, respectively (Li et al., 2019; Xu et al., 2022). The ANRIL and THRIL lncRNAs sponge miR-99a-5p, while the lncRNA HOXC-AS1 sponges miR-99a-3p (Chen et al., 2023b; Jiang et al., 2022b; Kotake et al., 2011; Liu et al., 2018).

Insulin stimulation and hypoxia are both related to HIF1α and may downregulate miR-99a and miR-100. Mechanistically, HIF1α activation suppresses miR-100 by promoting miR-100 sponged by lncRNA RAETIK (Blick et al., 2015; Blick et al., 2013; Chen et al., 2017a; Li et al., 2013b; Zhou et al., 2020). IGF1 and serum repress miR-99a expression via the phosphatidylinositol 3-kinase (PI3K) and mitogen-activated protein kinase (MAPK) kinase pathways (Yen et al., 2014). Similar inhibitory of miR-99a expression is induced by IFIT5, an IFN-induced protein (Huang et al., 2019). Ionizing radiation reduces the expression of miR-99b in pancreatic cancer cells and increases miR-99a and miR-100 expression in BC cells (Mueller, Sun & Dutta, 2013; Wei et al., 2013).

The functions of the miR-99 members can also be post-transcriptionally regulated via RNA editing. In humans, adenosine-to-inosine (A-to-I) conversion catalyzed by adenosine deaminase acting on RNA (ADAR) enzymes is the primary type of RNA editing. In miRNAs, the ADAR receptor regulates the processing of precursor miRNAs into mature miRNAs, and such processing affects the miRNA sequence, potentially altering its target genes and regulatory functions (Kawahara et al., 2007). A single A-to-I change at the -6 residue of primary miR-100 leads to enhanced miRNA processing by Drosha and consequently upregulates miR-100 both in vitro and in vivo (Chawla & Sokol, 2014). Wang et al. (2017) have identified A-to-I RNA editing hotspots in the miR-99a-5p mature sequence among 20 cancer types. In contrast, Tregs exhibit C-to-U RNA editing in the miR-100 seed region, which alters the miR-100 target from mTOR to SMAD2, further affecting Treg differentiation and formation (Negi et al., 2015).

MiRNA polymorphisms and gene fusions, which also influence the expression of miR-99 family members, are common in tumors. The fusion gene NRIP1-MIR99AHG, detected in acute myeloid leukemia (AML), results in the overexpression of miR-99a transcription and disruption of the tricistronic miR-99a/let-7c/miR-125b-2 cluster, facilitating the production of T-cell progenitors and accelerating leukemia progression (Kerbs et al., 2022). In acute lymphoblastic leukemia (ALL), a fusion occurs between the TEL (ETV6) gene (on chromosome 12) and the RUNX1 gene (on chromosome 21), thus upregulating members of the miR-99a/let-7c/miR-125b cluster and miR-100 (de Oliveira et al., 2012; Gefen et al., 2010). Single nucleotide polymorphisms (SNPs) in the transcription factor-binding sites of primary miRNAs result in the abnormal expression of mature miRNAs. For instance, the miR-100 SNP rs1834306 (T>C) reduces miR-100 expression, whereas rs1834306 (A>G) increases its expression, thus promoting Hirschsprung disease by directly or indirectly suppressing the functions of the associated pathways (Motawi et al., 2019; Zhu et al., 2020).

The regulation of miRNA expression is sex-dependent. For example, in BC and endometrial cancer, miR-100 expression is associated with the positivity of estrogen and progesterone receptors (Mattie et al., 2006; Zhou et al., 2010). Steroid hormones and their corresponding receptor agonists regulate the miR-99 family members. The androgen receptor agonist CI-4AS-1 reduces miR-100 and miR-125 expression in BC cells (Ahram et al., 2017). Androgen treatment activates nuclear translocation of the androgen receptor, which binds to AU-rich elements in the MIR99AHG enhancer and recruits the histone methyltransferase EZH2, thereby reducing the expression of MIR99AHG and its embedded miRNAs (Sun et al., 2014). This suggests that the regulation of miR-99 by androgens and their receptors is both transcriptional and epigenetic.

The regulation of mRNAs by miRNAs does not always occur simply via upstream or downstream regulation but sometimes happens in a feedback loop. IGF1 suppresses miR-99a expression, and its receptor, IGF1R, is a target of miR-99a-5p (Yen et al., 2014). TGF-β increases transcription of miR-99a, -99b, and -100 via SMAD2/3. However, SMAD2 is targeted by miR-99a/-100~125b tricistrons, implying negative feedback between TGF-β and miR-99 family members (Emmrich et al., 2014a; Ottaviani et al., 2018; Turcatel et al., 2012). NF-κB binds to the miR-100 promoter and directly activates miRNA transcription, and miR-100, in turn, activates NF-κB by targeting TRAF7 (Jeon et al., 2015). In ESCC, METTL14 mediates TRIB2 mRNA degradation via miR-99a-5p, whereas TRIB2 induces ubiquitin-mediated proteasomal degradation of METTL14 in a COP1-dependent manner (Liu et al., 2021b) (Fig. 1).

Involvement of the mir-99 family in cancer

Associations between miRNAs and malignancies have been widely examined, and miRNA dysregulation is involved in various cancers. Members of the miR-99 family have been reported to exhibit different effects in different cancer types. In particular, they may contribute to the initiation and progression of cancers, either as tumor-suppressive (TS) miRNAs or oncomiRs (Fig. 2, Table 1).

Figure 2 Regulation of cellular signaling pathways by miR-99 family.

miR-99 family regulates cellular activities by mediating various of signaling pathways, including TP53, MAPK, IGF, FGF, TNF and PI3K/AKT/mTOR. By targeting the crucial factors in these pathways, miR-99 family is able to modulate the phenotypes of cancer cells (cell cycle, proliferation, cell death, stemness, genotoxic resistance, inflammation, glycolysis, metabolic reprogramming, angiogenesis and VETC).

Table 1 Identified targets and regulatory effects of miR-99 family members in human cancers.

miR-99 family member	Expression of miR-99 family members	Target	Cancerous context	Effect	Reference	
miR-99a-3p	↓	STAMBP	HNSCC	Inhibition of migration and invasion	Okada et al. (2019)	
	↓	BMI1	GC	Cell apoptosis	Liu et al. (2018)	
	↓	MMP8	GC	Inhibition of proliferation	Jiang et al. (2022b)	
	↑	TRIM21	GC	Induction of proliferation, migration, invasion and EMT	He et al. (2024)	
	↓	RRM2	ccRCC	Inhibition of proliferation	Osako et al. (2019)	
	↓	GRP94	Papillary thyroid cancer	Inhibition of EMT, migration and invasion	Gao et al. (2021)	
	↓	NOVA1, DTL and RAB27B	IPA	Inhibition of cell growth and metastasis	Zhao et al. (2021)	
miR-99a-5p	↓	mTOR	OSCC	Inhibition of proliferation	Yan et al. (2012)	
	↓	mTOR	Lung cancer	Cell apoptosis, delaying cancer progression	Gu et al. (2013), Han et al. (2021), Oneyama et al. (2011), Song et al. (2014)	
	↓	mTOR	ESCC	Inhibition of proliferation	Sun et al. (2013)	
	↓	mTOR	BC	Cell apoptosis, inhibition of migration, invasion and sphere formation ability	Hu, Zhu & Tang (2014), Yang et al. (2014)	
	↓	mTOR	CRC	Inhibition of proliferation, invasion and migration	Zhu et al. (2019)	
	↓	mTOR	RCC	Inhibition of migration and invasion	Cui et al. (2012)	
	↓	mTOR	BCa	Inhibition of proliferation	Liu et al. (2019)	
	↓	cmTOR	Cervical cancer	Inhibition of proliferation and invasion	Wang et al. (2014a)	
	↓	dmTOR, IGF1R and FKBP51	ALL	Dexamethasone sensitivity	Li et al. (2013a)	
	↓	IGF1R	OSCC	Inhibition of migration, invasion and lung colonization	Yen et al. (2014)	
	↓	IGF1R	NSCLC	Inhibition of proliferation, migration and invasion	Chen et al. (2015), Chen et al. (2023b)	
	↓	IGF1R	RCC	Inhibition of cell growth	Sun et al. (2014)	
	↓	aIGF1R, mTOR	HNSCC	Inhibition of proliferation and migration	Chen et al. (2012)	
	↓	dIGF1R, mTOR and raptor	Adrenocortical cancer	Inhibition of proliferation	Doghman et al. (2010)	
	↓	MTMR3	Oral cancer	Inhibition of migration and invasion	Kuo et al. (2014)	
	↓	ICMT	OSCC	Inhibition of proliferation, migration, and invasion	Sun & Yan (2021)	
	↓	aFGFR3	Lung cancer	Inhibition of cell growth and metastasis	Jing et al. (2018)	
	↓	FGFR3, mTOR	Lung cancer	Inhibition of cell growth	Oneyama et al. (2011)	
	↓	FGFR3	BC	Inhibition proliferation, migration and invasion	Long et al. (2019)	
	↓	FGFR3	EOC	Inhibition of proliferation	Jiang et al. (2014)	
	↑ (Cisplatin-resistant vs. parental)	CAPNS1	GC	Cisplatin resistance	Zhang et al. (2016)	
	↓	AKT1	NSCLC	Inhibition of proliferation, migration and invasion	Yu et al. (2015)	
	↓	AKT1, mTOR	EC	Inhibition of proliferation and invasion	Li et al. (2016)	
	↓	HS3ST3B1	NSCLC	Inhibition of proliferation, migration and invasion	Zhai, Li & Lin (2024)	
	↓	NOX4	NSCLC	Inhibition of migration and invasion	Sun et al. (2016)	
	↓	E2F2, EMR2	Lung cancer	Inhibition of EMT and cancer stemness	Feliciano et al. (2017)	
	↓	bFAM64A	LUAD	Delaying cancer progression	Mizuno et al. (2020)	
	↓	bFAM64A	BC	Inhibition of migration and invasion	Shinden et al. (2021)	
	↑ (Doxorubicin-resistant vs. parental)	COX-2	BC	Doxorubicin sensitivity	Garrido‐Cano et al. (2022)	
	↓	CDC25A	BC	Cell cycle arrest	Qin & Liu (2019)	
	↓	CDC25A	Cervical cancer	Cell apoptosis	Gu & Bao (2022)	
	↑ (After RT vs. before RT)	SNF2H	BC	RT sensitivity	Mueller, Sun & Dutta, (2013)	
	↓	TRIB2	ESCC	RT sensitivity	Liu et al. (2021b)	
	↓	IGF1R	ESCC	Inhibition of proliferation, migration, invasion and EMT	Mei et al. (2017)	
	↓	IGF1R	Cholangiocarcinoma	Inhibition of migration, invasion and cancer stemness	Lin et al. (2016)	
	↓	IGF1R, mTOR	HCC	Inhibition of proliferation	Li et al. (2011)	
	↓	HOXA1	HCC	Inhibition of invasion and migration	Tao et al. (2019)	
	↑ (HSC vs. other hematopoietic cell populations)	HOXA1	AML	LSC self-renewal	Khalaj et al. (2017)	
	↓	aSMARCA5, SMARCD1, mTOR	PCa	Inhibition of cell growth	Sun et al. (2011)	
	↓ (Gemcitabine-resistant vs. parental)	SMARCD1	BCa	cell senescence	Tamai et al. (2022)	
	↓	RRAGD	Cervical cancer	Inhibition of invasion and migration	Wang et al. (2022a)	
	↓	CTDSPL, TRIB2	AML, CML	Induction of proliferation	Zhang et al. (2013)	
	↓	TNFAIP8	OSa	Cell cycle arrest	Xing & Ren (2016)	
miR-99b-3p	↓	GSK3β	OSCC	Inhibition of proliferation	He et al. (2015), Jakob et al. (2019)	
	↓	HoxD3	GC	Cell cycle arrest	Chang et al. (2019)	
	↓	PCDH19	HCC	Inhibition of proliferation, invasion and migration	Yao et al. (2019)	
	↓	NR6A1	PDAC	Inhibition of proliferation and invasion	Li et al. (2024)	
	↓	SRPK1	OC	Inhibition of viability	Xu et al. (2022)	
	↓ (paclitaxel-resistant vs. parental)	PPP2CA	BC	Induction of migration, proliferation, and paclitaxel sensitivity	Mao et al. (2024)	
	↓	CYLD	Melanoma	Cell apoptosis	La et al. (2020)	
miR-99b-5p	↓	FGFR3	NSCLC	Inhibition of cell growth and EMT	Du et al. (2018), Kang et al. (2012)	
	↓	FGFR3	CRC	Inhibition of proliferation, invasion and migration	Ning et al. (2023)	
	↑ (BRCA1wt vs. BRCA1mut)	TRAF2	BC	NF-κB pathway	Tanic et al. (2012)	
	↓	IGF1R	GC	Inhibition of proliferation	Wang et al. (2018b)	
	↓	IGF1R	PCa	Inhibition of proliferation, migration and invasion	Jiang et al. (2022a)	
	↑ (H. pylori+ vs H. pylori-)	mTOR	GC	Cell death	Yang, Li & Jia (2018)	
	↓ (After RT vs. before RT)	mTOR	PDAC	RT sensitivity	Wei et al. (2013)	
	↓	ARID3A	ESCC	Inhibition of invasion and migration	Ma et al. (2017)	
	↑ (Cisplatin-resistant vs. parental)	MTMR3	GC	Cisplatin sensitivity	Sun et al. (2020)	
	↓	CLDN11	HCC	Inhibition of invasion and migration	Yang et al. (2015)	
	↓ (MYCN-amplified vs. non-MYCN-amplified)	PHOX2B	NB	Doxorubicin sensitivity	Holliday et al. (2022)	
	↓	HS3ST3B1	BCa	Inhibition of proliferation, invasion	Li et al. (2019)	
miR-100-3p	↓	ATG10	NPC	Inhibition of proliferation and migration	Peng et al. (2022)	
	↑	LKB1	Head and neck Cancer	Promoting cancer progression	Figueroa-González et al. (2020)	
	↓	BMPR2	GC	Inhibition of cell growth	Peng et al. (2019)	
	↓	SNRPD1	HCC	Cell autophagy	Wang et al. (2022b)	
	↓	ErbB3	GBM	Inhibition of cell growth	Alrfaei et al. (2020)	
miR-100-5p	↓	HOXA1	NPC	Inhibition of proliferation	He et al. (2020)	
	↓	HOXA1	BC	Inhibition of cell motility	Chen et al. (2014)	
	↓	IGF1R	NPC	Inhibition of migration and invasion	Sun et al. (2018)	
	↓ (Cancer-associated fibroblasts-derived exosomes vs. normal fibroblasts derived exosomes)	IGF1R	ESCC	Inhibition of lymph angiogenesis	Chen et al. (2023a)	
	↓	IGF1R	Chordoma	Inhibition of proliferation and EMT	Zhang et al. (2020)	
	↓	PLK1	NPC	Radiation sensitivity	Shi et al. (2010)	
	↓	PLK1	HCC	Cell apoptosis	Chen, Zhao & Ma (2013)	
	↓	PLK1	Cervical cancer	Inhibition of proliferation	Peng et al. (2012)	
	↓	RASGRP3	NPC	Inhibition of proliferation and invasion	Peng et al. (2020)	
	↓	SMARCA5, SMRT	GBM	Inhibition of proliferation	Alrfaei et al. (2020), Alrfaei, Vemuganti & Kuo (2013)	
	↓	SMARC5A	NSCLC	Inhibition of migration and invasion	Li et al. (2021)	
	↓	SMARCA5	BC	Inhibition of EMT	Chen et al. (2014)	
	↓	ACKR3	NSCLC	Inhibition of brain metastasis	Ma et al. (2019), Zhang, Song & Zeng (2023)	
	↓	mTOR, PLK1, HOXA1	Lung cancer	Chemotherapy sensitivity	Feng, Wang & Chen (2012), Han et al. (2020a), Liu et al. (2012), Qin et al. (2017), Xiao et al. (2014).	
	↓	FOXA1, FZD8	BC	Inhibition of proliferation, migration and invasion	Jiang et al. (2016), Xie et al. (2021)	
	↓	FZD8	PTC	Inhibition of proliferation	Ma & Han (2022)	
	↑ (Trastuzumab-resistant vs. parental)	DLG5	BC	Trastuzumab sensitivity, inhibition of cancer stemness	Bai et al. (2022)	
	↓	IGF2	BC	Inhibition of proliferation	Gebeshuber & Martinez (2013)	
	↓	IGF2	HCC	Inhibition of cancer stemness	Seol et al. (2020)	
	↓	mTOR	ESCC	Inhibition of migration and invasion	Shi et al. (2019), Sun et al. (2013), Zhang et al. (2014b)	
	↓	mTOR	GC	Delaying cancer progression	Chen et al. (2018b)	
	↓	mTOR	CRC	Inhibition of proliferation, migration, invasion	Fujino et al. (2017), Jahangiri et al. (2022)	
	↓	mTOR	PCa	Inhibition of proliferation, migration and invasion	Ye, Li & Wang (2020)	
	↓	mTOR	BCa	Inhibition of proliferation and motility	Xu et al. (2013)	
	↓	mTOR	Cervical cancer	Inhibition of proliferation, migration and invasion	Yao et al. (2022)	
	↓	NOX4	RCC	Inhibition of proliferation, migration and invasion	Liu et al. (2022c)	
	↓	CXCR7	ESCC	Inhibition of migration and invasion	Zhou et al. (2016b)	
	↓	CXCR7	GC	Inhibition of cell growth	Cao et al. (2018)	
	↓	CXCR7	HCC	Inhibition of proliferation and invasion	Ge et al. (2021)	
	↓	ZBTB7A	GC	Inhibition of invasion and metastasis	Shi et al. (2015)	
	↑	HS3ST2	GC	Cisplatin sensitivity	Yang et al. (2015)	
	↑	RNF144B	GC	Promoting cancer progression	Yang et al. (2017)	
	↓	Lgr5	CRC	Inhibition of proliferation, migration and invasion	Zhou et al. (2015)	
	↓	RAP1B	CRC	Inhibition of cell growth and invasion	Peng et al. (2014)	
	↓	CLDN11	HCC	Inhibition of invasion and migration	Wang et al. (2023)	
	↓	AGO2	PCa	Inhibition of migration, invasion, EMT and cancer stemness	Wang et al. (2014b)	
	↓	FGFR3	PCa	Inhibition of proliferation, migration and invasion	Wu et al. (2015)	
	↓	FGFR3	GBM	Cisplatin sensitivity	Luan et al. (2015)	
	↓	SATB1	Cervical cancer	Inhibition of proliferation, migration and invasion	Huang et al. (2020)	
	↑	EPDR1	OC	Induction of migration and invasion	Liu et al. (2021a)	
	↓ (RT-resistant vs. sensitive)	ATM	GBM	RT sensitivity	Ng et al. (2010)	
	↑	ATM	AML	Inhibition of cell apoptosis	Sun, Wang & Luo (2020)	
	↑	RBSP3	AML	Inhibition of cell differentiation	Zheng et al. (2012)	
	↓ (After 131I exposure vs. before exposure)	RBSP3	Follicular thyroid carcinoma	Cell cycle arrest	Zhang et al. (2014a)	
Notes:

In the second column, those not indicated specifically were all meaning as tumor tissue vs. normal tissue.

a Common target of miR-99a-5p, -99b-5p, and -100-5p in the cited study.

b Common target of miR-99a-3p and -5p in the cited study.

c Common target of miR-99a-5p, -99b-5p in the cited study.

d Common target of miR-99a-5p, -100-5p in the cited study.

ESCC, esophageal squamous cell carcinoma; BC, breast cancer; IGF, insulin-like growth factor; NPC, nasopharyngeal carcinoma; AML, acute myeloid leukemia; ALL, acute lymphoblastic leukemia; HCC, hepatocellular carcinoma; CRC, colorectal cancer; PDAC, pancreatic ductal adenocarcinoma; OSCC, oral squamous cell carcinoma; HNSCC, head and neck squamous cell carcinoma; NSCLC, non-small cell lung cancer; LUAD, lung adenocarcinoma; EMT, epithelial–mesenchymal transition; GC, gastric cancer; VETCs, vessels that encapsulate tumour clusters; ccRCC, clear cell renal cell carcinoma; BCa, bladder cancer; PCa, prostate cancer; EOC, epithelial ovarian cancer; GBM, glioblastoma; NB, neuroblastoma; CML, chronic myeloid leukemia; Osa, osteosarcoma; HSC, hematopoietic stem cell; RT, radiation treatment.

Oral, head and neck cancer

Members of the miR-99 family play diverse roles in the development and progression of oral, head, and neck cancer. In oral squamous cell carcinoma (OSCC), the ectopic expression of miR-99b-3p suppresses the p65 (RelA) and G1 regulators (cyclin D1, CDK4, and CDK6) and inhibits cell proliferation by targeting glycogen synthase kinase-3β (GSK3β) (He et al., 2015). In OSCC, miR-99a-5p has been shown to target mTOR and impair cancer cell proliferation (Yan et al., 2012). Ectopic expression of miR-99a-5p markedly reduces cell migration and invasion by targeting IGF1R and myotubularin-related protein 3 (MTMR3) (Kuo et al., 2014; Yen et al., 2014). Suppression of IGF1R expression by miR-99a-5p reduces lung colonization by oral cancer cells in vivo. Isoprenyl cysteine carboxymethyltransferase (ICMT), responsible for enhanced invasiveness, is a target of miR-99a-5p (Sun & Yan, 2021).

miR-100-5p inhibits the proliferation of nasopharyngeal carcinoma by targeting HOXA1 (He et al., 2020). miR-100-5p represses migratory and invasive abilities by targeting IGF1R and RASGRP3 (Peng et al., 2020; Sun et al., 2018). miR-100-5p targets polo-like kinase 1 (PLK1) to reduce CDC25C levels, thereby enhancing the cytotoxicity of radiation treatment (RT) (Shi et al., 2010). miR-100-3p targets ATG10 and activates the PI3K/AKT signaling pathway to attenuate autophagy, which leads to the suppression of proliferation and migration (Peng et al., 2022).

Various studies have reported the multifaceted roles of miR-99a in head and neck squamous cell carcinoma (HNSCC). miR-99a-5p, miR-99b-5p, and miR-100-5p commonly target both IGF1R and mTOR, thereby inducing cell proliferation and migration and enhancing apoptosis (Chen et al., 2012). STAMBP, targeted by miR-99a-3p, facilitates the migration and invasiveness of HNSCC cells (Okada et al., 2019). The levels of miR100-3p (targeting the tumor suppressor LKB1) and miR-100-5p are elevated in HNSCC tissues (Figueroa-González et al., 2020).

Lung cancer

Members of the miR-99 family are frequently downregulated in lung cancer (Feliciano et al., 2017; Han et al., 2021; Mizuno et al., 2020; Ye et al., 2023). miR-99a-5p, miR-99b-5p, and miR-100-5p co-target FGFR3 and sequentially repress Erk1/2 and AKT, thereby delaying lung cancer progression (Du et al., 2018; Jing et al., 2018; Kang et al., 2012; Oneyama et al., 2011). miR-99a-5p induces apoptosis and inhibits proliferation, migration and invasion in non-small cell lung cancer (NSCLC) by targeting IGF1R, AKT1 and HS3ST3B1 (Chen et al., 2015; Chen et al., 2023b; Yu et al., 2015; Zhai, Li & Lin, 2024). Moreover, miR-99a-5p reduces ROS accumulation by targeting NOX4 and inhibits the MAPK pathway by targeting mTOR, which is also targeted by miR-99a-5p in lung adenocarcinoma (LUAD) (Gu et al., 2013; Oneyama et al., 2011; Song et al., 2014; Sun et al., 2016). miR-99a-5p targets the oncogenic proteins E2F2 and EMR2, thereby preventing epithelial-mesenchymal transition (EMT) and repressing pluripotency in the cancer stem-like cell (CSC) population (Feliciano et al., 2017). In LUAD tissue, the potential anti-tumor gene MIR99AHG and its embedded miRNAs are downregulated, partly owing to the copy number deletion of MIR99AHG. Consequently, miR-99a-5p and MIR99AHG synergistically promoted autophagy and delayed LUAD progression by targeting mTOR (Han et al., 2021). Another oncogenic target, FAM64A, is regulated by miR-99a-5p and miR-99a-3p (Mizuno et al., 2020).

miR-100-5p exhibits a tumor-suppressive function similar to that in NSCLC. It is under-expressed and consequently no longer suppresses SMARCA5, a gene that promotes cell invasion (Li et al., 2021). Similarly, miR-100-5p inhibits lung-cancer-derived brain metastasis by downregulating ACKR3 and blocks EMT and Wnt/β-catenin signaling by targeting HOXA1 (Han et al., 2020a; Ma et al., 2019; Zhang, Song & Zeng, 2023). Conversely, SCLC cell lines resistant to cisplatin, etoposide, and adriamycin exhibit upregulated miR-100-5p expression, which maintains the resistant phenotype by targeting HOXA1 to prevent apoptosis and cell cycle arrest (Han et al., 2020a; Xiao et al., 2014). In cisplatin-resistant NSCLC cells, miR-100-5p is downregulated, and mTOR is upregulated (Qin et al., 2017). In docetaxel-resistant LUAD cells, miR-100-5p inhibits PLK1 to modulate resistance (Feng, Wang & Chen, 2012); this also in NSCLC, miR-100-5p targets PLK1 to attenuate growth, arrest the G2/M phase, and enhance apoptosis (Liu et al., 2012). miR-100-5p is upregulated in TNF-related apoptosis-inducing ligand (TRAIL) NSCLC cells. In conjunction with miR-21 and miR-30c, miR-100-5p further strengthens NF-κB signaling, establishing a positive-feedback loop that leads to TRAIL resistance and EMT, thus promoting cell survival after TRAIL treatment (Jeon et al., 2015). miR-100-5p confers resistance to ALK tyrosine kinase inhibitors in EML4-ALK NSCLC cells (Lai et al., 2019). However, many drug resistance mechanisms of miR-100-5p have not yet been examined, and further research is needed.

Breast cancer

Due to its high heterogeneity, the diagnosis of BC remains challenging. MiR-99a-5p induces apoptosis, inhibits cell migration and invasion, and reduces sphere formation by targeting mTOR and FGFR3 (Hu, Zhu & Tang, 2014; Long et al., 2019; Yang et al., 2014). Moreover, it reduces the ability of the ATP-binding cassette subfamily G member 2 (ABCG2) to perform doxorubicin efflux and elevates the sensitivity to doxorubicin by targeting COX-2 (Qin & Liu, 2019). In BC, miR-99a-3p and miR-99a-5p co-target FAM64A, affecting cell migration and invasion (Shinden et al., 2021). Both miR-99a-5p and miR-100-5p are induced after irradiation, further validating that miR-99a-5p reduces SNF2H expression and prevents BRCA1 recruitment to sites of DNA damage (Mueller, Sun & Dutta, 2013). BRCA1 reactivation induces the transcription of miR-99a and miR-99b, which then target TRAF2, a key regulator of the NF-κB and MAPK pathways (Tanic et al., 2012). In contrast, based on the genomic profiling of BC tissues, miR-99b also functions as an oncomiR in BC and is associated with high homologous recombination deficiencies and intra-tumor heterogeneity (Oshi et al., 2022). miR-99b-3p is highly expressed in paclitaxel-resistant BC cells and induces cell migration and paclitaxel resistance by targeting PPP2CA directly or promoting M2 polarization of macrophages (Mao et al., 2024).

miR-100 is commonly downregulated in BC owing to the hypermethylation of its host gene, MIR100HG. In BC, miR-100-5p represses proliferation, migration, and invasion by targeting FOXA1 and FZD8, abolishes trastuzumab resistance and CSC-like properties by targeting DLG5, inhibits proliferation and induces apoptosis by targeting IGF2, affects EMT and suppresses tumorigenesis, cell motility, and invasiveness by targeting SMARCA5 and HOXA1 (Bai et al., 2022; Chen et al., 2014; Gebeshuber & Martinez, 2013; Jiang et al., 2016; Xie et al., 2021).

The expression of miR-100 is related to estrogen and progesterone receptor positivity. Ectopic expression of miR-100 promotes luminal differentiation and renders basal-like BC stem cells responsive to hormonal therapy (Mattie et al., 2006). In the luminal-A BC population receiving adjuvant endocrine therapy, miR-100 levels were positively correlated with better overall survival (OS). Further molecular analysis revealed that miR-100 expression was inversely correlated with the expression of various genes, including PLK1, FOXA1, mTOR, and IGF1R, which are involved in resistance to hormonal therapy (Pidíkova, Reis & Herichova, 2020).

Esophageal cancer

ESCC is the predominant subtype of esophageal cancer, accounting for approximately 90% of all esophageal cancer cases worldwide. MiR-99a-5p blocks CSC persistence and sensitizes ESCC cells to radiotherapy by first targeting TRIB2 and blocking HDAC2 activation via the mTOR signaling pathway (Liu et al., 2021b; Sun et al., 2013). Similar to most cancer types, IGF1R is targeted by miR-99a-5p and miR-100-5p in ESCC, resulting in the suppression of tumor cell proliferation, migration, invasion, and SLUG-induced EMT (Chen et al., 2023a; Mei et al., 2017). miR-100-5p prevents cell migration and invasion by targeting mTOR and CXCR7 (Shi et al., 2019; Sun et al., 2013; Zhang et al., 2014a; Zhou et al., 2016b). miR-99b-5p represses ARID3A (a target of miR-125a and let-7e) to inhibit cell invasion and migration (Ma et al., 2017).

NGS-based profiling has revealed that miR-99a is upregulated in cisplatin-resistant esophageal cancer cells (Pandey et al., 2023). In patients with esophageal adenocarcinoma, high miR-100-3p expression predicts poorer survival, and patients who fail to achieve a pathological complete response have higher miR-99b expression (Feber et al., 2011; Skinner et al., 2014). Thus, the role of miR-99 family members in esophageal cancer may be subtype-dependent.

Gastric cancer

miR-99 is dysregulated in gastric cancer (GC). miR-99a-3p acts as a TS miRNA in GC, inducing apoptosis by targeting BMI1 and preventing proliferation by targeting MMP8 (Jiang et al., 2022b; Liu et al., 2018). miR-99b-3p and -5p induce cell cycle arrest in the S phase by targeting HoxD3 and IGF1R, respectively (Chang et al., 2019; Wang et al., 2018b). miR-100-3p represses tumor growth by targeting BMPR2, whereas miR-100-5p inhibits cell growth by targeting CXCR7, represses invasion and metastasis by targeting ZBTB7A (Shi et al., 2015), and activates the autophagic pathway by targeting mTOR (Cao et al., 2018; Chen et al., 2018b; Peng et al., 2019).

One of the leading causes of GC is Helicobacter pylori infection, which is associated with the expression of the miR-99 family. In cancer tissues, miR-99b-3p and -5p levels are higher when H. pylori is present (Chang et al., 2015). miR-99b-5p induces GC cell death via autophagy by targeting mTOR and eliminating intracellular H. pylori (Yang, Li & Jia, 2018).

Although miR-99 members counteract GC, they also promote its progression. In GC with poor prognosis, miR-99a-3p is overexpressed and promotes cell proliferation, migration, invasiveness, and EMT by targeting TRIM21 (He et al., 2024). Similarly, in cisplatin-resistant GC cells, miR-99a-5p is upregulated, whereas its target gene calpain small subunit 1 (CAPNS1) is downregulated. Silencing miR-99a-5p activates the catalytic subunits of CAPNS1 (calpain1 and calpain2), leading to GC cell apoptosis (Zhang et al., 2016). In cisplatin-resistant GC cells, high expression of miR-99a 5p silences MTMR3, which fails to repress autophagy, thus maintaining resistance (Sun et al., 2020). Yang et al. (2015, 2017) demonstrated the oncogenic effects of miR-100-5p. They targeted HS3ST2 to inhibit the Notch signaling pathway and apoptosis, attenuating cisplatin sensitivity. Upregulation of miR-100-5p is associated with primary tumorigenesis and progression of GC; miR-100-5p targets RNF144B, an E3 ubiquitin ligase. RNF144B interacts with pirh2, another p53 E3 ubiquitin ligase, to accelerate ubiquitin-mediated p53 degradation (Yang et al., 2017, 2015). This finding suggests that miR-100-5p is associated with genomic instability (Xu et al., 2023).

Colorectal cancer

The miR-99 family plays an important role in intestinal cancer. miR-99a-5p and miR-99b-5p impair the proliferation, invasion, and migration of intestinal cancer cells by targeting mTOR and FGFR3 (Ning et al., 2023; Zhu et al., 2019). In CRC cells, miR-100-5p targets mTOR, contributing to their proliferation, migration, and invasion of CRC cells (Fujino et al., 2017; Jahangiri et al., 2022). miR-100-5p has been identified to target Lgr5 and RAP1B (Peng et al., 2014; Zhou et al., 2015). miR-100-5p, miR-125b, and their host, MIR100HG, are overexpressed in cetuximab-resistant CRC and HNSCC cells (Liu et al., 2022a). Lu et al. (2017) further examined the potential mechanism, reporting that miR-100 and miR-125b coordinately repress five Wnt/β-catenin negative regulators, resulting in increased Wnt signaling. In contrast, Wnt inhibition restored cetuximab responsiveness in cetuximab-resistant cells. Interestingly, miR-99a-5p was associated with blood sugar levels. Advanced glycation end-products (AGEs) effectively reduce miR-99a-5p levels in vitro. In CRC tissue, the expression of miR-99a-5p was lower in patients with diabetes mellitus (DM) than in those without DM (Zhu et al., 2019). It has been found that insulin downregulates miR-99a-5p (Li et al., 2013b). Therefore, it is worth examining the risk of using insulin for glycemic control in patients with CRC and DM.

Hepatocellular carcinoma

Dysregulation of the miR-99 family has been linked to hepatocellular malignancies. The expression of miR-99a-5p in cancerous liver tissues was lower than in normal tissues. miR-99a-5p suppresses the invasion and migration of HCC cells by targeting HOXA1 (Tao et al., 2019). Furthermore, miR-99a-5p induces G1/S arrest and suppresses cell proliferation by targeting IGF1R and mTOR (Li et al., 2011). Specifically, the activation of mTOR induces PKM2 and HIF-1α expression, subsequently promoting glucose consumption and lactate production and thus promoting glycolysis (Li et al., 2013b). miR-100-5p exhibits effects similar to those of miR-99a-5p and functions as a TS miRNA in HCC cells. The dysregulation of miR-100 and PLK1 is closely associated with carcinogenesis, and miR-100-5p targets PLK1 to reduce HCC growth and enhance apoptosis (Chen, Zhao & Ma, 2013; Petrelli et al., 2012). miR-100-5p targets IGF2 to repress the AKT/mTOR pathway, thereby abolishing the maintenance of CSC properties and attenuating invasive and proliferative abilities by targeting CXCR7 (Ge et al., 2021; Seol et al., 2020). Angiopoietin 2 (Angpt2), essential for forming vessels encapsulating tumor clusters (VETCs), facilitates the entry of the entire tumor cluster into the bloodstream in an invasion-independent manner. miR-100-5p reduces the protein levels of Angpt2 by blocking the mTOR-p70S6K pathway, thereby decreasing VETC formation and metastasis (Zhou et al., 2016a). miR-100-3p reduces mTOR levels, thereby triggering autophagy by targeting SNRPD1 (Wang et al., 2022b). Both miR-99a and miR-100 target mTOR, suggesting that mTOR-autophagy signaling is a core pathway targeted by the miR-99 family.

Although miR-99a and miR-100 exhibit suppressive functions in HCC, miR-99b is oncogenic (Yang et al., 2015; Yao et al., 2019). miR-99b-3p induces proliferation, invasion, and migration by targeting PCDH19, whereas miR-99b-5p and miR-100-5p promote invasion and migration by targeting CLDN11 (Wang et al., 2023; Yang et al., 2015; Yao et al., 2019).

Pancreatic adenocarcinoma

There are multifaceted findings regarding the role of miR-100-5p in PDAC; some studies have reported that miR-100-5p is downregulated in cancerous tissues, whereas others have noted the opposite (Dobre et al., 2021; Panarelli et al., 2012). Contrary to the findings for miR-99a-5p in CRC, the expression of miR-100-5p is higher in patients with PDAC and DM than in those without DM, and the expression of miR-100-5p may be associated with high HbA1c (Hara et al., 2023). The levels of miR-100-5p and E-cadherin are negatively correlated, implying that miR-100-5p reduces overall survival by inducing robust EMT and motility (Hara et al., 2023; Ottaviani et al., 2018).

miR-99a, miR-100, and miR-125b are upregulated in gemcitabine-resistant PDAC cells (Dhayat et al., 2015). The impairment of miR-100 or miR-125b activity can reduce CSC marker expression and sensitize cells to gemcitabine treatment (Ottaviani et al., 2018). Similarly, the miR-99b/let-7e/miR-125a cluster inhibited cell proliferation, invasion, and metastasis by targeting NR6A1 (Li et al., 2024). In PDAC, miR-99b-5p induces radiation resistance by targeting mTOR (Wei et al., 2013).

Urological cancer

The role of the miR-99 family in genitourinary cancers depends on the tumor subtype and progression stage. In clear cell renal cell carcinoma (ccRCC), miR-99a is overexpressed (Oliveira et al., 2017); however, its expression is reduced in bladder cancer (BCa) and prostate cancer (PCa) (Borkowska et al., 2023; Sun et al., 2011). miR-100-5p, which represses migration, invasion, EMT, and stemness by targeting AGO2, is under expressed in PCa, whereas it promotes migration and prevents apoptosis in RCC (Chen et al., 2017b; Wang et al., 2014b). Interestingly, although miR-100 expression decreased during the transition from localized to metastatic PCa, biochemical recurrence was associated with high levels of miR-100. This discrepancy suggests that miR-100 is a context-dependent miRNA, sometimes acting as an oncomiR and sometimes as a TS miRNA (Leite et al., 2011). While miR-100 is expressed as a biomarker in all representative lesions during carcinogenesis in PCa, it exhibits progressive downregulation during the progression from precancerous to advanced metastatic cancer (Leite et al., 2013). In patients with bladder urothelial carcinoma, miR-100-5p shows heterogeneous expression and its role depends on the cancer stage. Dip et al. (2012) suggested that under expression of miR-100 in low-grade pTa specimens may increase FGFR3 levels, thus facilitating mutations by increasing cell turnover and the selection of mutant cells. In contrast, in invasive tumors, miR-100 overexpression may induce THAP-2 silencing and lead to the dysregulation of cell proliferation (Dip et al., 2012).

miR-99a-5p suppresses growth, migration, and invasion by targeting mTOR in RCC and bladder cancer and FGFR3 in PCa, and inhibits tumor growth by targeting IGF1R (Cui et al., 2012; Liu et al., 2019; Sun et al., 2014; Wu et al., 2015). miR-100-5p and miR-99b-5p exhibit similar antitumor effects by targeting mTOR, NOX4, HS3ST3B1, and IGF1R (Jiang et al., 2022a; Li et al., 2019; Liu et al., 2022c; Xu et al., 2013; Ye, Li & Wang, 2020).

In PCa, an androgen analog has been found to repress miR-99a-5p and miR-100-5p expression, thereby partly reducing the tumor-suppressive effects of the miR-99 family; furthermore, the miR-99 family inhibits androgen-receptor activity by targeting SMARCA5, SMARCD1, and mTOR, thus reducing prostate-specific antigen levels. However, inhibiting androgen-independent cell growth by the miR-99 family requires the presence of an androgen receptor (Sun et al., 2011). miR-99b-5p inhibits AR-mediated mTOR translocation and reduces mTOR expression, enhancing docetaxel-induced cytotoxicity (Gujrati et al., 2022). Likewise, miR-99a-5p inhibits the expression and nuclear translocation of mTOR, SMARCD1, and AR, and miR-99a-5p participates in metabolic reprogramming by recruiting the AR/mTOR complex to its target genes, thereby inhibiting EMT-mediated metastasis and elevating the cytotoxicity of enzalutamide (Enz) and abiraterone acetate (Waseem, Gujrati & Wang, 2023; Waseem & Wang, 2024). Androgen deprivation induces the expression of miR-100-5p, which is necessary for the survival and proliferation of PCa cells in a hormone-independent manner (Nabavi et al., 2017).

The expression of miR-99a-5p in gemcitabine-resistant BCa cells is lower than that in parental BCa cells, and its ectopic expression induces cellular senescence by targeting SMARCD1, thereby restoring sensitivity to gemcitabine (Tamai et al., 2022). In sunitinib-resistant ccRCC cells, miR-99a-3p is similar in targeting RRM2 to repress proliferation and induce apoptosis (Osako et al., 2019).

Gynecological cancers

Cervical, ovarian, and endometrial cancers are common gynecological tumors that impose a significant burden on women. In endometrial cancer, miR-99a-5p induces apoptosis and represses cancer cell proliferation and invasion via dual suppression of AKT and mTOR (Li et al., 2016). In cervical cancer, miR-99a-5p enhances apoptosis, represses glycolysis by targeting RRAGD, suppresses migration and invasion, and promotes apoptosis by targeting CDC25A (Gu & Bao, 2022; Wang et al., 2022a). In epithelial ovarian cancer (EOC), miR-99a-5p inhibits proliferation by targeting FGFR3 (Jiang et al., 2014). In cervical cancer, miR-99a-5p and miR-99b-5p suppress cell proliferation and invasion by targeting mTOR (Wang et al., 2014a). miR-99b-3p inhibits ovarian cancer cell viability by targeting SRPK1 (Xu et al., 2022).

Low miR-100-5p expression in cervical cancer is associated with unfavorable clinical outcomes (Yao et al., 2022). miR-100-5p inhibits proliferation by targeting PLK1 and represses migration and invasion by targeting SATB1 and mTOR (Huang et al., 2020; Peng et al., 2012; Yao et al., 2022). Consistently, in cisplatin-resistant EOC cells, miR-100-5p is downregulated, and re-expression of miR-100-5p reduces mTOR and PLK1 protein levels and induces apoptosis and cell cycle arrest in the G1 phase (Guo et al., 2016). In contrast, miR-100-5p expression is elevated in ovarian cancer, which promotes cell migration and invasion by targeting EPDR1 (Liu et al., 2021a).

EOC-derived exosomes increased fibronectin and vitronectin expression by transporting miR-99a-5p in human peritoneal mesothelial cells. Treating EOC cells with HPMCs promoted their invasiveness, suggesting that cancer cell-derived exosomal miRNAs modulate the tumor microenvironment (TME) to provide feedback to the cancer phenotype (Yoshimura et al., 2018).

Neurological cancers

miR-99 family members are essential for reducing the malignant phenotype of glioblastoma (GBM). Overexpression of miR-99a inhibits FGFR3 and PI3K/AKT signaling, thereby augmenting the repression of proliferation and induction of apoptosis via photofrin-based photodynamic therapy (Chakrabarti, Banik & Ray, 2013). The tumorigenic FGFR3–TACC3 fusion has been consistently detected in GBM. This fusion promotes cell proliferation and tumor progression by allowing GBM cells to escape recognition by miR-99a-5p (Parker et al., 2013). Studies examining the anti-GBM mechanism of miR-100 revealed that the miR-100-5p guide strand represses STAT3 by targeting SMARCA5 and SMRT, whereas miR-100-3p reduces AKT and ERK phosphorylation by targeting ErbB3 (Alrfaei et al., 2020; Alrfaei, Vemuganti & Kuo, 2013). miR-100-5p sensitizes GBM cells to ionizing radiation by targeting ATM, repressing their growth and migration, and enhancing their chemosensitivity by targeting FGFR3 (Luan et al., 2015; Ng et al., 2010). Neuroblastoma (NB) is a malignant tumor derived from immature neuronal cells of the sympathetic nervous system that is frequently reported in children. In NB, miR-99b-5p acts as a chemosensitizing miRNA and enhances DOX cytotoxicity by targeting PHOX2B (Holliday et al., 2022).

Hematological malignancies

In leukemia, regulating the miR-99 family and its clusters is complex and specific to the leukemia type. miR-99a, miR-100, and let-7 negatively regulate pro-proliferative genes, partially counteracting the hyperproliferation of hematopoietic stem cells induced by miR-125b, thereby conferring a steady-state growth advantage and preventing the exhaustion of miR-125b-transduced hematopoietic stem cells (Emmrich et al., 2014a). In AML, miR-100-5p inhibits apoptosis by targeting ATM, arrests the differentiation of human granulocytes and monocytes, and promotes cell survival by targeting RBSP3 (Sun et al., 2020; Zheng et al., 2012). miR-99a-5p inhibits differentiation of hematopoietic and AML stem cells, promoting self-renewal by targeting HOXA1. It promotes proliferation and inhibits apoptosis in AML and chronic myeloid leukemia (CML) cells by targeting CTDSPL and TRIB2 (Khalaj et al., 2017; Zhang et al., 2013). Contrary to their role as oncomiRs in AML, miR-99a-5p, and miR-100-5p were downregulated in childhood ALL tissues, especially in high-risk groups. Furthermore, they activate glucocorticoid receptors to increase dexamethasone sensitivity and suppress the IGF1R/mTOR pathway to induce apoptosis by targeting FKBP51, IGF1R, and mTOR (Li et al., 2013a). In AML, miR-99b and miR-125a induce proliferation and maintain LSC function (Uebbing et al., 2021).

In diffuse large B-cell lymphoma, miR-100-5p functions as a TS miRNA and suppresses the proliferation, migration, and invasion of cancer cells (Shu et al., 2022). Multiple myeloma (MM), the most prevalent malignant plasma cell disease, is characterized by the abnormal proliferation of bone marrow plasma cells. Wei et al. (2023) detected the upregulation of miR-100-5p in MM tissues and found that the inhibition of miR-100-5p induced targets, such as CLDN11, ICMT, MTMR3, RASGRP3, and SMARCA5, thus reducing metastasis and inducing apoptosis.

Other cancers

In thyroid cancer tissues, especially in the advanced stage, miR-100-5p r suppresses proliferation, induces apoptosis, and inactivates Wnt/β-catenin signaling by targeting FZD8 and RBSP3; moreover, it is under expressed in these cancers (Ma & Han, 2022; Zhang et al., 2014a). By targeting GRP94, miR-99a-3p disrupts anoikis resistance and inhibits the cytoplasmic relocation of ITGA2, thereby suppressing EMT (Gao et al., 2021). In cholangiocarcinoma in vivo, the miR-99a/let-7c/miR-125b cluster reduces STAT3 activity and further suppresses migration and invasiveness, as well as suppressing CSC-like mammosphere generation and tumorigenicity by targeting IGF1R and IL-6, respectively (Lin et al., 2016).

In melanoma, miR-99b-3p targets CYLD, thereby preventing RIPK1 polyubiquitination and inducing apoptosis (La et al., 2020). The levels of miR-99a/-100 are significantly higher in nevi than in malignant lesions and are negatively associated with IGF1R expression; restoration of miR-99a/-100 reduces IGF1R expression and melanoma cell proliferation (Damsky et al., 2015). Notably, IGF1R levels were lower in PTEN-silenced cells than in CDKN2A-silenced cells, indicating that IGF1R depletion by miR-100 was greater with PTEN silencing than with CDKN2A silencing, which is consistent with other findings (Majewska et al., 2022).

In osteosarcoma, miR-99a-5p induces cell death and cell cycle arrest by targeting TNFAIP8 (Xing & Ren, 2016). Chordoma is a malignant mesenchymal tissue bone tumor in which miR-100-5p represses proliferation and EMT and induces apoptosis by targeting IGF1R (Zhang et al., 2020). In epithelial cells, miR-100-5p targets HOXA1, thus reducing BCL-2 expression and inducing apoptosis. Similarly, in cutaneous squamous cell carcinoma cells, inhibition of miR-100 enhances radiation resistance (Fahim Golestaneh et al., 2019).

In invasive pituitary adenomas, miR-99a-3p is under expressed, and its expression is negatively correlated with invasiveness. The ectopic expression of miR-99a-3p inhibits cell growth, metastasis, and tube formation in endothelial cells by targeting NOVA1, DTL, and RAB27B (Zhao et al., 2021). miR-99a-5p and miR-100-5p are downregulated in childhood adrenocortical tumors and repress the proliferation of both adrenocortical tumors and pediatric adrenocortical carcinoma cells by targeting mTOR, RAPTOR, and IGF1R (Doghman et al., 2010).

Roles in cancer diagnosis, prognosis, therapeutic-response prediction, and treatment

miRNAs have various advantages as cancer biomarkers. As previously discussed, the expression of the miR-99 family members varies among different cancers and during malignancy. The expression of this miRNA is associated with progression from normal to precancerous lesions and from early to advanced-stage cancer. Consequently, miR-99 expression may predict cancer onset, act as a prognostic marker, and predict therapeutic responses in cancer-derived tissues and possibly in serum.

Roles in diagnosis and progression

miRNAs have been reported to play crucial roles in tumorigenesis and development, and their expression is associated with clinical outcomes. Their roles as potential diagnostic and prognostic biomarkers in various cancers have also been examined. The expression of miR-99 varies between cancerous and normal tissues and sera. Consequently, miR-99 members have been verified as biomarkers for cancer diagnosis, individually and as components of miRNA signatures (Holubekova et al., 2020; Kaba et al., 2023). Interestingly, tissues and serum often exhibit contrasting miRNA expression patterns. The expression of miR-99a-5p was significantly lower in BC tissues than in healthy tissues, while the opposite pattern was observed in the plasma of these patients (Garrido-Cano et al., 2020). Receiver operating characteristic (ROC) curve analysis revealed that miR-99a-5p has good diagnostic potential, even for detecting early BC, suggesting that circulating miR-99a-5p is a novel and promising non-invasive biomarker for BC detection. However, miRNA signatures comprising several miRNAs appear more accurate than those comprising individual miRNAs. For example, miRNA signatures can be used to classify endometrial cancer tissue (miR-99a/-100/-199b) and plasma (miR-99a/-199b) samples with a higher accuracy than single miRNAs (Torres et al., 2012). In addition to being measured in plasma, miRNAs can be measured in other body fluids, such as urine, for urological cancer diagnosis (Liu et al., 2024; Pospisilova et al., 2016). Salido-Guadarrama et al. (2016) identified a miR‑100/200b signature that discriminates between patients with PCa and benign hyperplasia, achieving greater accuracy than prostate-specific antigen.

The expression of miR-100 is lower in ESCC tissues than in non-cancerous esophageal tissues, and its dysregulation is associated with an advanced clinical stage, distant metastasis, increased depth of invasion, and poor survival probability, suggesting that miR-100 could serve as a biomarker for ESCC prognosis (Zhou et al., 2014). In GC, miR-100 expression increases with the progression stage, making it a marker of tumor advancement (Ueda et al., 2010). Similarly, miR-100-5p is a marker of tumorigenic progression in cervical cancer, and its expression decreases with progression from low-grade cervical intraepithelial neoplasia (CIN) to high-grade CIN and then to cancer (Li et al., 2011). In patients with OSCC, serum miR-99a-5p levels are higher after tumor resection than before (Chen et al., 2018a). Conversely, in patients with ESCC, serum miR-100 levels are lower after surgery, suggesting that miR-99 levels reflect tumor burden and serve as an auxiliary indicator of surgical effectiveness (Wu et al., 2014). Recurrence, another important indicator of cancer progression, has been reported to be associated with the miR-99 family. MiR-99a is highly expressed in pediatric AML and CML at the time of diagnosis and relapse; however, its expression is significantly reduced during complete remission (Zhang et al., 2013).

RNA editing of miR-99a/-99b occurs more frequently in cancers (most cancers) than in normal tissues and is correlated with survival probability (Pinto et al., 2018). For example, patients with LUAD with a loss of A-to-I miR-99a-5p editing exhibit reduced overall survival (Maemura et al., 2018). miR-99a editing is associated with different molecular drivers and signaling pathways in different cancers, such as the TP53 pathway in BC and HNSCC and the HRAS and NRAS pathways in thyroid carcinoma (Wang et al., 2017).

Overall, the miR-99 members exhibit substantial potential as diagnostic and prognostic biomarkers. They can be used to distinguish between cancerous and non-cancerous tissues, sera, and other body fluids of patients with malignant tumors and the normal population. In particular, circulating miR-99 members are attractive markers for early non-invasive cancer detection and can be easily analyzed in large batches of clinical samples.

Roles in predicting therapeutic responses

miR-99 expression varies dynamically during anticancer therapy, and its dysregulation may reverse treatment efficiency. Therefore, its expression is potentially helpful in predicting the treatment response. MiR-99b-5p is downregulated in imatinib-resistant CML patients. Among patients with high-risk myelodysplastic syndromes or AML with myelodysplasia-related changes, miR-100-5p levels are higher in azacitidine-responsive patients than in non-responders (Krejcik et al., 2018, Yap et al., 2017). In patients with BC, serum miR-100-5p expression was significantly lower in those who responded to initial dovitinib treatment than in those with treatment-resistant metastatic BC. This helps clinicians decide whether to continue planned treatment (Shivapurkar et al., 2017).

MiRNA signatures display impressive response-predictive ability. For example, a signature comprising circulating miR-100, miR-92a, miR-16, miR-30e, miR-144-5p, and let-7i can distinguish patients with oxaliplatin-based chemo resistant CRC from those with chemo sensitive CRC (Han et al., 2020b). The circulating miR-21/-99b/-375 panel is an effective indicator of the preoperative chemoradiotherapy response in locally advanced rectal cancer (Campayo et al., 2018). In esophageal adenocarcinoma, miR-99b and three other miRNAs form a signature that predicts a pathological complete response to neoadjuvant chemoradiation, suggesting the potential of miR-99 members as indicators of neoadjuvant chemotherapy efficacy (Skinner et al., 2014). Among patients with melanoma, higher miR-100-5p expression predicts greater clinical benefit from PD-1-inhibitor treatment, whereas the opposite has been reported for miR-100 in myeloid-derived suppressor cells (Huber et al., 2018; Sloane et al., 2021). In BCa tissues, miR-100-5p expression is inversely correlated with that of PD-L1 and PD-L2 (El Ahanidi et al., 2021). Although miR-100 plays a role in immunotherapy, this requires further validation in future studies.

In summary, the association between miR-99 expression and treatment response can be used to predict resistance. Quantifying miR-99 levels in patients with malignant lesions provides a sensitive, effective, and timely method for detecting resistance, guiding the selection of chemotherapy regimens, and improving prognosis.

Clinical translation potential of the miR-99 family

As our understanding of miRNAs in cancer has improved, they have emerged as attractive tools and targets for novel therapeutic approaches. Two key strategies for miRNA-based cancer therapy are: (1) administration of synthetically derived miRNA mimics to restore the activity of mutated or deleted TS miRNAs or (2) inhibition of endogenous oncomiRs. Considering their simple structure and ease of synthesis, miRNAs exhibit strong competitiveness as treatment options, potentially bypassing expensive medicinal chemistry research.

Many studies have suggested the potential therapeutic use of miR-99 family members, the reintroduction of their synthetic mimics, or the use of their antisense sequences. miRNAs are unable to cross cell surface membranes directly. Extracellular vesicles, particularly exosomes, provide a key pathway for the delivery of extracellular miRNAs into recipient cells, where they alter their genetic profile (Garcia-Martin et al., 2022). Connections between lung cancer cell membranes and mast cells induce the release of extracellular vesicles enriched with miR-100-5p and miR-125b, resulting in accelerated lung cancer cell proliferation (Shemesh et al., 2023). Exosomal miR‑100‑5p from highly invasive HCC cells promotes the migration and invasion of low-invasive HCC cells, further confirming the role of exosomes in miRNA transport (Wang et al., 2023). In contrast, MSC-derived exosomal miR-100 efficiently suppressed CRC cell proliferation and induced apoptosis by reducing the expression of mTOR, cyclin D1, KRAS, and HK2 (Jahangiri et al., 2022).

Synthetically derived miRNAs are rapidly degraded by the plasma. Delivery systems that enhance in vivo delivery and minimize miRNA degradation during systemic circulation have been proposed for the clinical translation of miRNAs. As an effective theranostic antitumor approach, nanomaterials provide an efficient platform for loading miRNAs. Sun et al. (2017) designed and synthesized a nanovector for miR-100 delivery that predominantly targeted and suppressed FGFR3, thereby significantly inhibiting the growth of FGFR3-amplified patient-derived xenografts. Holliday et al. (2022) constructed nanoparticle complex-modified miR-99b-5p mimics with MYCN amplification that increased the susceptibility of neuroblastoma patient-derived xenografts with MYCN amplification. Nanoparticles that deliver miR-99a-5p along with doxorubicin or anti-vascular endothelial growth factor (VEGF) antibody to tumor cells have been successfully developed, and treatment with nanoparticle-loaded combinations was more effective and less toxic than treatment with free doxorubicin or VEGF antibody alone (Cai et al., 2017; Garrido‐Cano et al., 2022). The intranasal nanoparticle-mediated co-delivery of miR-100 and antisense (anti)-miR-21 bypasses the blood-brain barrier and potentiates the effects of systemic temozolomide treatment in GBM (Sukumar et al., 2019). In mice, intranasally delivered nanoparticles carrying tumor-suppressive genes (thymidine kinase, p53, and nitroreductase) along with therapeutic miRNAs (anti-miR-21, anti-miR-10b, and miR-100) predominantly accumulated in the lungs, thus reducing triple-negative BC–lung metastases (Liu et al., 2022b).

In addition to functioning directly in tumor cells, miRNAs can also affect the TME, potentially regulating cancer progression and providing another strategy for cancer treatment. Tumor-associated macrophages (TAMs), the primary immune components of the TME, mediate various tumor-promoting mechanisms such as angiogenesis stimulation, tumor migration enhancement, and antitumor immunity suppression. In BC, high miR-100 expression maintains the TAM phenotype by targeting mTOR and increasing IL-1ra secretion via stat5a-mediated transcriptional regulation, thus enhancing metastasis, stemness, chemoresistance, and features of malignancy (Wang et al., 2018a). In contrast, miR-99b induces the conversion of TAMs into an antitumor phenotype with enhanced immune surveillance. When conjugated to a nucleic acid drug delivery system and then delivered into TAMs, miR-99b promotes M1 macrophage polarization, thus enhancing phagocytosis and antigen presentation by targeting κB-Ras2 or mTOR. Furthermore, it suppresses M2 macrophage polarization by repressing the mTOR/IRF4 axis, suppressing tumor growth in HCC and Lewis lung cancer. Amplification of the M1-like effect by miR-99b overexpression in TAMs causes tumor regression by reprogramming the antitumor immune microenvironment (Wang et al., 2020).

Conclusion

Increasing evidence has revealed that miR-99 family members are crucial in diverse cellular processes, as well as in disease development and progression, particularly in cancer. Notably, the same miR-99 family members have been reported to play different roles in different cancer types, leading to conflicting opinions regarding the role of the miR-99 family in cancer. Some of these discrepancies may result from experimental differences and require further validation, whereas others may be related to the expression and status of the target genes. Combined with advanced computational algorithms to analyze large-scale genomic and transcriptomic data, this could contribute to predicting the binding sites of miR-99 on target genes more accurately or identifying previously unknown targets. Moreover, the addition of computational biology may be helpful for a deeper understanding of complex regulatory mechanisms.

Currently, most research on the miR-99 family is limited to the role of miRNA and target genes and subsequent changes in cell physiological activities regulated by target genes. Exploring the detailed molecular docking between miR-99 family members and their target genes provides crucial information about the structural basis of their interactions, making it possible to identify off-target docking site mutations in advance, and design small-molecule modulators to regulate the miR-99 target interactions. In the realm of animal studies, more sophisticated animal models that closely mimic human cancer could be developed to study the roles of miR-99 family members in a more clinically relevant context, such as patient-derived xenograft (PDX) models. In addition, new biomaterials loaded with agents and miRNAs can be synthesized and applied in animal models to observe their antitumor effects and accompanying immune responses.

Most cancer types exhibit dysregulated miR-99 expression. Consequently, this family has the potential to serve as diagnostic and prognostic markers for malignancies. Many studies have revealed that circulating miRNAs in human serum and other body fluids can be utilized as biomarkers for cancer, thus enabling clinicians to perform non-invasive analyses. Considering that miR-99 expression is affected by chemotherapy, radiation treatment, and other anticancer therapeutic methods, it may also serve as a predictive biomarker for therapeutic responses. The tumor-suppressive or tumor-promoting properties of miRNAs have been widely reported, and many studies have revealed the therapeutic roles of miR-99. Nonetheless, further research is required before the miR-99 family can be used in clinical applications.

Supplemental Information

Supplemental Information 1 Summary of the article selection or exclusion criteria used to construct this review.

Additional Information and Declarations

Competing Interests

The authors declare that they have no competing interests.

Author Contributions

Yueyuan Wang conceived and designed the experiments, performed the experiments, analyzed the data, prepared figures and/or tables, authored or reviewed drafts of the article, and approved the final draft.

Dan Huang performed the experiments, authored or reviewed drafts of the article, and approved the final draft.

Mingxi Li analyzed the data, prepared figures and/or tables, and approved the final draft.

Ming Yang conceived and designed the experiments, prepared figures and/or tables, authored or reviewed drafts of the article, and approved the final draft.

Data Availability

The following information was supplied regarding data availability:

This is a literature review.

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
