# Peer review of "MicroRNA-99 family in cancer: molecular mechanisms for clinical applications"

_PeerJ, doi:10.7717/peerj.19188_

## Round 0.1 · original submission · Major Revisions

I offer you a major revision. However, as reviewer 2 pointed out, this manuscript has problems with similarity to the published paper (https://doi.org/10.1002/wrna.1635). Upon resubmission of the revision, please make sure that your paper demonstrates a different perspective than the published paper. In addition, please provide appropriate citations for the previously published work.

·

Basic reporting

No comment

Experimental design

I would suggest the authors to include a flowchart on the survey methodology by stating the selection criteria and the number of studies from each database.

Validity of the findings

The authors should carefully discuss about the multifaceted roles of miR-99 family members, instead of just label them as "conflicting roles". As discussed in the manuscript, some of the miR-99 family members act on different targets to induce different effects, either promoting tumour progression or anti-cancer, in a specific cancer type. Please choose your wordings carefully when interpret the findings from previous literature.

As for Conclusion, please rewrite the first paragraph, with regards to my comments above. The author should propose some suggestions for future direction, e.g computational biology, molecular docking and animal studies. This part is just too superficial in the Conclusion

Additional comments

I notice that the authors use semicolons extensively to connect the ideas and sentences throughout the whole manuscript, which may create grammatical errors and confusion for the readers. Please improve on this matter. I also suggest the authors to carefully review the whole manuscript and choose the proper wordings in the context of cancer studies. For example, in Table 1, I came across "postponement of cancer progression", which is an uncommon term to see in cancer research.

As for Table 1, the authors should include the expression levels of miR-99 family members in each reported study. Please standardised the in-text citation format for Table 1. I would appreciate if the authors could have explained the abbreviations of cancer types and some terms (RT, LSC, VETC) in the "effect" column in the table footnote.

I also suggest the authors to carefully review Figure 2 and reduce the number of wordings inside the figure as the figure caption is sufficient to explain everything.

Reviewer 3 ·

Basic reporting

1. The review is generally written in clear English, however, there are some points, phrases and sentences that can be hard to understand. There are many typos in the manuscript as well. These make the text unclear for non-native readers. Therefore, I suggest an entire language check and proofreading to correct the typos.
For example;
• There are unclear phrases in line 36: ‘30% of the human gene set’; line 76: ‘little relevance’
• The sentences in lines 93-96 are not connected for fluency: ‘miRNA families are groups of homologous genes that share highly similar seed sequences but encode different mature sequences. They include three homologues miR-99a, miR-99b, and miR-100 encoded on chromosomes 21, 19, and 11, respectively, and modulated by different host genes’.
• The sentence in lines 231-232 is not understandable: ‘miRNAs are enriched in exosomes, which can be taken up by neighboring or distant cells, which they then modulate.’
• Typos: line 59: ‘con-served’, line 120: ‘ex-pression’, line 213: ‘in vitro and in vivo’ should be italicised.
2. The references used in the entire text are mostly outdated studies. More recently published papers can be used in general.
3. Professional article structure, figures, and tables are informative. However, they highly resemble the structure, figures and tables in this review: Eniafe J, Jiang S. MicroRNA-99 family in cancer and immunity. WIREs RNA. 2021; 12:e1635. https://doi.org/10.1002/wrna.1635
4. This review has several common points with the recently published review cited below.
Eniafe J, Jiang S. MicroRNA-99 family in cancer and immunity. WIREs RNA. 2021; 12:e1635. https://doi.org/10.1002/wrna.1635
5. The introduction is quite enough for the reader.

Experimental design

Survey methodology is not detailed enough to replicate the study. The including and excluding criteria are also not notified. There is a phrase as ‘little relevance’ for exclusion criteria.
The heading ‘The audience this review is intended for’ can be unnecessary to use for the fluency of the manuscript.
The most of the sources are adequately cited. There are some informative sentences which do not have a reference such as the sentence in lines 96-100.

Validity of the findings

The findings are well discussed and connected to the main goal.

Additional comments

The study has an impact on cancer research. However, the manuscript can be more specific for novelty. As a general idea, I comment on the authors to language editing.

---

## Round 0.2 · Minor Revisions

The reviewers acknowledge the improvement of your manuscript. However, they still find some issues that need to be addressed. I hope that these issues can be addressed.

·

Basic reporting

In my opinion, the authors should remove the paragraph 2 in Introduction (line 47 to 61) on the miRNA synthesis which is not relevant to introduce the subject. The authors should discuss more on the burden of cancers worldwide and address the shortcomings of current diagnosis and treatment, which lead to the ongoing efforts to discover potential biomarkers.

The literature references provided are sufficient.

This manuscript provides a good review of current research progress on miR-99 family.

I would also suggest to remove Figure 1 which is irrelevant to the main text of this manuscript

Experimental design

No comment

Validity of the findings

In my opinion, section 3 Regulation of miR-99 expression is too lengthy and lack of organisation, I would suggest the authors to refine the whole section to be more precise.

Additional comments

Please standardise the referencing style throughout the whole manuscript, like citing the authors at the end of sentences.

---

## Round 0.3 · Minor Revisions

The reviewers have accepted most of your revisions, but still require minor revisions. Please revise according to the reviewers' comments. Please be especially careful with your choice of words.

I have a question: Was this paper produced with the aid of LLM AI? Please answer this question when submitting the revised manuscript. If so, you must declare that (as per the journal's policies).

·

Basic reporting

No comment

Experimental design

No comment

Validity of the findings

No comment

Additional comments

Line 47 "The global cancer burden is astonishing" Please rewrite this into "The global cancer burden is increasing yearly" Astonishing is not a suitable word in this context

---

## Round 0.4 · accepted · Accept

I confirmed your revisions and now accept your manuscript for publication in PeerJ.